# Metabolomic Signatures of Alzheimer’s Disease Indicate Brain Region-Specific Neurodegenerative Progression

**DOI:** 10.3390/ijms241914769

**Published:** 2023-09-30

**Authors:** Mirela Ambeskovic, Giselle Hopkins, Tanzi Hoover, Jeffrey T. Joseph, Tony Montina, Gerlinde A. S. Metz

**Affiliations:** 1Canadian Centre for Behavioural Neuroscience, Department of Neuroscience, University of Lethbridge, Lethbridge, AB T1K 3M4, Canada; mirela.ambeskovic@uleth.ca (M.A.); giselletiede@gmail.com (G.H.); hoovtd@uleth.ca (T.H.); 2Hotchkiss Brain Institute, Cumming School of Medicine, University of Calgary, Calgary, AB T2N 1N4, Canada; jtjoseph@ucalgary.ca; 3Department of Chemistry and Biochemistry, University of Lethbridge, Lethbridge, AB T1K 3M4, Canada; 4Southern Alberta Genome Sciences Centre, University of Lethbridge, Lethbridge, AB T1K 3M4, Canada

**Keywords:** proton nuclear magnetic resonance (^1^H NMR) spectroscopy, aging, neuropathology, neurodegenerative disease, cognitive function, phenylalanine, GABA, energy metabolism, neurotransmission, mitochondrial dysfunction, oxidative stress, glucose metabolism

## Abstract

Pathological mechanisms contributing to Alzheimer’s disease (AD) are still elusive. Here, we identified the metabolic signatures of AD in human post-mortem brains. Using ^1^H NMR spectroscopy and an untargeted metabolomics approach, we identified (1) metabolomic profiles of AD and age-matched healthy subjects in post-mortem brain tissue, and (2) region-common and region-unique metabolome alterations and biochemical pathways across eight brain regions revealed that BA9 was the most affected. Phenylalanine and phosphorylcholine were mainly downregulated, suggesting altered neurotransmitter synthesis. N-acetylaspartate and GABA were upregulated in most regions, suggesting higher inhibitory activity in neural circuits. Other region-common metabolic pathways indicated impaired mitochondrial function and energy metabolism, while region-unique pathways indicated oxidative stress and altered immune responses. Importantly, AD caused metabolic changes in brain regions with less well-documented pathological alterations that suggest degenerative progression. The findings provide a new understanding of the biochemical mechanisms of AD and guide biomarker discovery for personalized risk prediction and diagnosis.

## 1. Introduction

The elderly population is growing worldwide and so are the concerns over the rise in aging-associated neurodegenerative diseases, such as Alzheimer’s disease (AD) [1,2]. AD is the most common neurodegenerative disease and affects approximately 55 million people worldwide, a number that is estimated to nearly triple by 2050 [1,2]. Common symptoms of AD include learning and memory deficits; emotional dysregulation; impaired abstract reasoning, language, and speech ability; and motor impairments [3,4,5]. These clinical symptoms are consequences of long-term complex brain changes, including inflammation, oxidative stress, and the accumulation of amyloid-beta and tau proteins, ultimately leading to neuropathological changes [6,7,8,9,10,11].

The formation of neurofibrillary tangles (NFTs) and amyloid-beta plaques (Aβ) in the brain is considered a key pathological hallmark of AD [11,12] associated with the atrophy of multiple brain regions [13]. The NFTs are initially observed in the hippocampus (HPC), then reach the entire brain via fugal movement, ultimately contributing to neuronal atrophy [2,5,14]. Aβ plaques are initially observed in the HPC and the cortex, such as frontal cortex (BA9), and later spread to other brain regions [15]. These pathological markers move across the brain, along with inflammation, oxidative stress, and impaired energy metabolism, leading to neuronal cell death [5,16].

Genetic risk factors, such as inheritance of apolipoprotein E (APOE), presenilin-1 (PSEN1), presenilin-2 (PSEN2), and amyloid precursor protein (APP) genes, can increase the risk of neuropathological changes linked to AD [4,6,12]. APOE is essential for the movement of fats in the bloodstream [4,5]. The alterations in PSEN 1 and 2 and APP can lead to the abnormal production of proteins and amyloid and lead to the formation of plaques [12], the hallmarks of AD. These genetic risk factors are associated with metabolic changes in the brain, including altered levels of lipids and low Aβ concentrations [12,14]. Although much is known about AD-related neuropathology and genetic risk factors, the underlying biochemical processes contributing to its onset and progression are still incompletely understood [5,17]. Thus, robust biomarkers of AD with early diagnostic and prognostic potential are still not available. Investigating AD-related metabolic changes across multiple brain tissues can critically advance the understanding of biochemical changes at the cellular level and potentially provide targets for novel predictive biomarkers of this disease.

Metabolomics is a powerful approach used to capture the complex dynamic metabolic changes associated with AD [14,18,19,20,21]. Metabolic fingerprints indicate the disrupted intercellular communication underlying the progressive pathological mechanisms of AD [22]. As metabolites are downstream from genomic, epigenomic, and physiological regulation, they provide robust indicators of neurodegenerative cellular processes [23,24]. AD-related metabolomics of post-mortem brain tissues has revealed metabolic changes in the HPC, frontal cortex [including Brodmann area (BA) 9], and temporal cortex [3,14,23,25,26]. The profiling of post-mortem AD tissues via proton nuclear magnetic resonance (^1^H NMR) spectroscopy demonstrated N-acetylaspartate (NAA) reduction in the grey matter of BA17 and superior temporal gyrus (BA22), among other regions [27,28]. However, most studies used targeted metabolomics and focused on specific brain areas, hindering effective biomarker discovery due to a lack of gross metabolic exploration across multiple brain regions [16,18,19,20,21,22,23].

In this study, we used an untargeted metabolomics approach to determine comprehensive metabolomic fingerprints linked to AD across eight brain regions [BA9, BA17, BA22, BA24, BA40, dentate nucleus (DN), HPC, and pons base (PB)] in post-mortem human tissues (Figure 1). The identification of region-specific versus common metabolites and biochemical pathways highlighted the functional connectivity between regions and revealed profiles of pathological progression in AD. The results provide targets for new potential predictive, diagnostic, and prognostic biomarkers of AD for personalized medicine approaches.

## 2. Results

### 2.1. Metabolomic Signatures of AD Occur across Brain Regions

This study involved post-mortem brain tissues from 11 AD patients and 11 healthy subjects (HS), including 5 females and 6 males in each group. The average age was 80 years old for AD participants and 65 years old for HS. The average neuropathology ABC score was 7.63 out of possible 9 in AD and 1.63/9 in HS subjects (Table 1).

AD significantly altered the metabolic profiles of all eight brain regions (BA9, BA17, BA22, BA24, BA40, DN, HPC, and PB; Figure 2; Table 2). Mann-Whitney (MW) and variable importance analysis based on random variable combination (VIAVC) best subset bins revealed the most (118) significantly altered metabolites in BA9 and the least (24) altered metabolites in BA24. HPC, BA22, BA40, and BA17 had 48, 41, 38, and 34 significantly altered metabolites, respectively (Table 2). Fewer metabolites were significantly altered in DN, with 27 metabolites changed, compared to 26 in PB (Figure 2A,B). These results demonstrate the most significant metabolic changes in brain regions where AD initially begins, such as BA9, HPC, BA22, and BA40. This finding is consistent with the neuropathological disease progression of AD and clinical observations of cognitive impairment [4,8,29]. Moreover, the present study demonstrates that AD leads to metabolic changes in brain regions with less well-documented pathological alterations (PB, DN and BA24).

Although AD altered the metabolomic profiles of each investigated brain region, the direction of the regulation varied between regions. Predominantly upregulated metabolites were observed in BA9, DN, and HPC. Approximately 90% of metabolites were upregulated in BA9, 63% in DN, and 60% in HPC. In contrast, mostly downregulated metabolites were found in BA40 (87%), BA24 (70%), PD (65%), BA17, and BA22, with 60% of metabolites being changed in AD tissues (Figure 2C,D). BA9 and HPC showed the most upregulated metabolites and are known to be the first and most affected by AD.

### 2.2. Biomarkers of AD: Brain Region Common and Unique Metabolites

When AD-associated metabolite alterations were compared across all investigated brain regions, some common and unique metabolites were identified. Common metabolites were identified in the hallmark regions most affected by AD (BA9, HPC) and the least affected regions (DN and PB) (Figure 3A). One common metabolite altered in the most and the least AD-affected regions was glycyl-glycine (Appendix A). Across regions involved in functional communication between the cortex and cerebellum (BA9, BA40, and DN), four common metabolites were identified (ethanol, serine, acetylcholine, and glycolic acid) (Figure 3B; Appendix A). For clinically relevant regions (BA9, BA17, and HPC) eleven common metabolites were identified (NAA, 2-amino-3-phosphonopronic acid, gamma-aminobutyric acid (GABA), leucine, phenylalanine, serine, tyrosine, valine, phosphorylcholine, creatine, and glycerol and malonic acid) (Figure 3C; Appendix A). Interestingly, NAA, GABA, leucine, glycerol, and malonic acid were all upregulated in BA9, BA17, and HPC. The least altered brain regions (BA24, DN, and PB) showed three common metabolites (isoleucine, gylcyl-glycine, and citric acid; Figure 3D; Appendix A). In all three brain regions, isoleucine and glycyl-glycine were downregulated, while citric acid was upregulated. When parietal and frontal regions (most affected by AD) were compared, nine common metabolites were identified (NAA, 2-amino-3-phosphonopronic acid, GABA, glucose, leucine, phenylalanine, serine, tyrosine, and valine and phosphorylcholine; Figure 3E; Appendix A). In all four regions, NAA and GABA were upregulated, while leucine was downregulated. The direction of the regulation for other metabolites varied.

VIAVC analysis identified common and unique metabolites that may serve as biomarkers of AD (Table 3). Corresponding metabolites with altered regulation in most brain regions included phenylalanine, NAA, GABA, phosphorylcholine, serine, and citric acid (Figure 3). Specifically, phenylalanine and phosphorylcholine were downregulated in all regions, except for their upregulation in BA9, while no changes were observed in DN and PB for phosphorylcholine (Figure 3F,L). Moreover, NAA and GABA were upregulated in all regions, except for DN and PB, where they were unchanged (Figure 3K,G). Arginine and serine were downregulated in BA17 and BA40, while serine was upregulated in BA9 and BA22 (Figure 3N,I). Citric acid and succinic acid were upregulated in BA9 and DN, while citric acid was upregulated in all other regions, except BA17 and BA40 (Figure 3J,M). Uniquely, lactic acid was downregulated in BA9 and HPC, while myoinositol was upregulated in BA17, HPC, and PB (Figure 3O,H). The metabolomic profiles, therefore, identified common metabolites altered across multiple brain regions as potential biomarkers of AD. These specific markers may be used for early disease diagnosis or prognosis from peripheral tissues such as blood. Possible biomarkers of metabolomics in AD are summarized in Table 3.

### 2.3. Pathological Pathway Discovery: AD Impacts Neurotransmission, Energy Metabolism, and Mitochondrial Regulation

Pathway topology analysis was carried out for each of the eight brain regions (Figure 4A), and only pathways with a *p*-value < 0.05 and impact factor higher than 0.1 were considered. Notable pathways affected by AD included alanine aspartate and glutamate (changed in all regions except BA17 and PB), phenylalanine metabolism and phenylalanine tyrosine, and tryptophan biosynthesis (all altered, except DN and BA24). Pyruvate metabolism and tricarboxylic acid (TCA) cycle was changed in BA9 and HPC and BA9, BA22, and DN, respectively. Glycerophospholipid metabolism was significantly altered in BA17, BA24, and BA40. Pathways uniquely altered in specific brain regions included D-glutamine D-glutamate in HPC, cysteine and methionine metabolism in BA22, glycolysis/gluconeogenesis, and the synthesis and degradation of ketone bodies in BA9 (Figure 4A).

Identified sub-biochemical pathways belonged to four super pathways, namely amino acids, energy, carbohydrates, and lipids. Interestingly, most of the pathways belonged to the amino acid sub-pathway, while lipids and carbohydrates were second most affected and energy was the least affected by TCA cycle changes (Figure 4B–D).

## 3. Discussion

AD is a devastating neurodegenerative disease with no cure and limited clinical diagnostic tools. Diagnosis usually occurs years after the neuropathological changes were initiated. Slow disease progression and multiple biochemical mechanisms involved in AD make the discovery of robust predictive biomarkers and treatments difficult. Metabolomics is a promising technique that can enhance the understanding of the biochemical mechanisms involved in the disease and potentially provide new biomarkers of AD. In the present study, an untargeted ^1^H NMR metabolomics and machine learning were used to profile human post-mortem brain tissues from eight regions to identify region-common and -unique metabolites and metabolic pathways involved in AD. The results demonstrate that AD has widespread consequences and impacts all of the investigated brain regions. Commonly altered metabolites are involved in metabolic energy regulation, neurotransmission, and mitochondrial function.

Previous experimental [14,30,31] and clinical studies in urine [24,32], blood [33], cerebrospinal fluid (CSF) [14,16], and brain tissues [18,22,26,34] indicate that AD is associated with altered metabolomic regulation. The present data support these observations and, for the first time, provide evidence of AD-related metabolomic changes across eight brain regions. The most affected brain region was BA9, a cortical region involved in higher cognitive functions and among the most severely affected by AD neuropathology [4,35]. Although neuropathological changes in BA17, BA24, DN, and PB are less commonly observed, the present data demonstrate profound AD-associated changes in these regions. For example, of the 27 altered metabolites in DN, 63% were upregulated, even though previous research showed little or no AD-related pathology in DN [36]. These changes may be functionally meaningful, considering the DN functions as the main projection between the cerebellum and the cortex [36]. Moreover, major altered metabolites were upregulated in BA9 and HPC, two regions also connected to the DN [37,38]. Importantly, citric acid and serine, which are metabolites necessary for energy metabolism, were part of the VIAVC Best Subset bins in the DN. This finding suggests that the DN may serve a compensatory role in the cerebellar regions to ensure functional connectivity to cortical regions and adjust for dysfunctional energy metabolism in cortical regions in AD [36,38].

AD is recognized as a disease of synaptic dysfunction and impaired neurotransmission with altered neuronal communication, which is central to the characteristic cognitive and emotional loss [3,35,39,40]. In the present study, AD significantly upregulated GABA and NAA levels in all regions, except for DN and PB. This finding confirms previous observations of a major alteration in the inhibitory neurotransmitter GABA in the frontal, temporal, and parietal cortexes and hippocampi of AD subjects [40,41,42,43]. For the first time, we report GABA alterations in primary visual cortex (BA17), BA22, and BA24. Moreover, this study reports increases in NAA, which is an amino acid that is essential for neuronal functioning [23,44,45,46], contradicting previous observations of NAA reduction in BA17, BA22, and HPC [47]. Nevertheless, NAA upregulation points to a potential compensatory increase in synthesis to offset the consequences of neuronal loss [27,47]. Here, GABA and NAA were corresponding metabolites to the VIAVC best subset bins (Table 2), which highlights the potential of these metabolites as biomarkers of AD. In addition, our data support previous evidence of glutamate upregulation in BA22 and HPC [28,48,49] and provide new evidence of altered glutamate levels in the DN and PB of AD subjects. Increased synaptic glutamate in AD pathology may lead to excitotoxicity by activating N-methyl-D-aspartic acid receptors (NMDAR), resulting in calcium overload [15,50]. Furthermore, NMDAR function may be further affected by the observed changes in AD-induced serine levels in all brain regions, except for BA24 and PB.

Phenylalanine and tyrosine are important precursors of catecholamine neurotransmitter synthesis, which is essential for normal cognitive function [33,51,52,53]. In this study, we report altered phenylalanine and tyrosine levels in all brain regions, except in BA24 and DN of AD patients, compared to healthy controls. Moreover, altered phenylalanine and tyrosine levels significantly impacted phenylalanine metabolism and phenylalanine, tyrosine, and tryptophan biosynthesis pathways. These data support previous studies that reported altered levels of phenylalanine and tyrosine in various AD tissues [24,54,55] and provide evidence of AD-affected brains having reduced levels of both phenylalanine and tyrosine [56]. Here, both phenylalanine and tyrosine were downregulated in most brain regions, except for BA9. Phenylalanine and tyrosine must be ingested from food and serve as precursors of catecholamine [33]; therefore, their depletion may contribute to impaired catecholamine synthesis, low catecholamine levels, and behavioural deficits, including cognitive and emotional impairments [21,53,57,58]. Then, it may be suggested that elevated levels of phenylalanine and tyrosine measured in prefrontal cortex (BA9) may serve as a compensatory mechanism, as this is where most cognitive processing takes place [58]. Moreover, previously increased phenylalanine and tyrosine levels were observed in chronic inflammatory conditions in association with neuropsychiatric symptoms [33,59], and such immune activation may represent critical factors involved in the pathogenesis of AD [60]. In this study, phenylalanine was identified as a corresponding metabolite to the VIAVC Best Subset bins and represents an excellent biomarker of AD, as altered phenylalanine concentrations are related to altered AD-associated catecholamine synthesis and immune activation.

Links between AD pathology, oxidative stress, and inflammation are well documented [61,62] and suggest that AD is characterized by changes in the arginine–proline and glycerophospholipid metabolisms [19,22,26,63]. Oxidative stress and neuroinflammation contribute to mitochondrial dysfunction reported in AD [64]. In this study, we report AD-associated changes in arginine–proline metabolism and glycerophospholipid metabolism in three brain regions (BA17, BA22, and BA40). The main metabolites involved in these pathways, namely arginine and phosphorylcholine, were corresponding metabolites to the VIAVC Best Subset bins (Table 3), which highlights their potential as effective biomarkers of AD.

Previously, mitochondrial dysfunction in AD was observed in association with alanine, aspartate, and glutamate metabolism. Paglia et al. (2016) showed that AD altered this pathway in the frontal cortex (BA9), with upregulation in glutamate, GABA, NAA, and citrate metabolites. In this study, alanine, aspartate, and glutamate metabolisms were altered in all brain regions, except BA17 and PB. Therefore, these findings, for the first time, show changes in alanine, aspartate, and glutamate metabolisms throughout the brain, indicating that these metabolites may represent early biomarkers of AD. Alterations in these metabolites contribute not only to mitochondrial dysfunction, but also form compounds involved in the TCA cycle; glycolysis; and glycine, serine, and threonine metabolisms. Ultimately, their alteration may indicate an impaired energy metabolism, along with reduced mitochondrial efficacy, a hallmark neuropathological feature of AD [19,28,41].

Similar to energy metabolism alterations in AD [5,19,65], the present pathway enrichment analyses revealed that AD potentially altered energy metabolism by commonly affecting multiple pathways in multiple brain regions. Notably, glycine, serine, and threonine metabolism pathways were altered in all regions, except BA24, BA40, and PB. The TCA cycle was altered in BA9, BA22, and DN, while pyruvate metabolism was altered in BA9 and HPC. The TCA cycle is essential to adenosine triphosphate (ATP) production, and its alteration may be reflective of reduced energy production in AD [63]. Metabolites that play essential roles in the TCA and pyruvate cycles, including citric acid, lactic acid and succinic acid, were identified as part of the VIAVC best subset (Table 3) in this study.

This study provides evidence that AD is associated with brain-wide metabolic changes. Some of the same metabolites, including GABA, arginine, serine, phosphorylcholine, and phenylalanine, were previously identified in the CSF of AD patients [14,16,17,57]. CSF contains the richest amounts of metabolites found in the brain, and it has been established that CSF and blood show a significant overlap in terms of metabolites among AD patients [14]. Therefore, it may be suggested that the metabolic changes observed in this study represent potential biomarkers of AD in peripheral tissues such as blood. However, it is important to mention limitations of this study, including the (1) effects of the post-mortem interval (larger interval in AD patients), (2) effects of the often “slow dying” of dementia patients compared to the “fast dying” of the controls (most controls rapidly died from cardiac disease, while AD patients often slowly died from pneumonia or a urinary track infection; slow death might be associated with increased hypoxia and, hence, increased lactic acid; increased gluconeogenesis; and decreased TCA function), and (3) effects on aging alone, since the AD group in general is older than the control group (Table 1). Nonetheless, the identification of common and unique metabolites in multiple brain regions highlights their potential as biomarkers of AD.

## 4. Materials and Methods

### 4.1. Participants and Demographic Information

The participants were Alberta, Canada, residents who lived either at home or in a nursing home or were admitted to hospital. The basic demographics, including age, gender, post-mortem delay (PMD), and the pathological characterization of samples, are shown in Table 1. Ante mortem written informed consent forms received from the donor or next of kin indicated that all donations were voluntary. The ethically approved use of tissues in scientific research was permitted. All protocols were approved by the University of Calgary Hotchkiss Brain Institute (AB, Canada, approval # REB14-0452_REN5) and the University of Lethbridge (AB, Canada, approval # 2021-038).

### 4.2. Tissue Samples, Collection and Classification

Post-mortem brain tissues were obtained between 2015 and 2018. A total of 22 post-mortem human brain samples were obtained from the Calgary Brain Bank (AD, n = 11, 5 females, 6 males, ages 80 ± 9 yrs.; healthy subjects (HS), n = 11, 5 females, 6 males, ages 65 ± 12 yrs.; Table 1). The brain tissues were collected from eight brain regions, including superior prefrontal gyrus (superior prefrontal gyrus or Brodmann Area 9 (BA9), primary visual cortex (BA17), superior temporal gyrus (BA22), anterior cingulate gyrus (BA24), inferior parietal lobule (BA40), dentate nucleus (DN), hippocampus (HPC), and pons base (PB; see Figure 1).

AD brain tissues were collected from patients in the late stages of the disease. Brains were collected and processed according to the standardized operational protocol of human brain banking in Canada. In brief, within 18–55 h post-mortem, brains were collected, and hemispheres were dissected sagittal [4]. At random, one of the brain hemispheres was freshly collected, and specific regions were dissected and stored in −80 °C. The other hemisphere was fixed in 10% formalin for a week, then the tissues were dissected and paraffin-embedded for neuropathological examination or ABC score (Amyloid and Braak and Braak tau distribution and CEDAR score) assessment [4].

ABC score assessments were completed by a registered neuropathologist. In brief, the tissues were characterized as AD or control based on assessment for neuritic plaques and neurofibrillary tangles, as established by the Consortium to Establish a Registry for Alzheimer’s Disease (CEDAR) [29], amyloid plague distribution and Braak tau stage [4,7]. Categories were ranked on a scale from zero to three, with three being most deviated from typical non-diseased tissue characteristics. If the summation of each of these rankings was greater than five, the tissue was classified as AD (Table 1).

### 4.3. Tissue Preparation

Ultrafiltration was used to extract water-soluble metabolites from each tissue sample. In brief, tissue samples were removed from −80 °C storage and kept on ice to thaw at room temperature. Approximately 150 mg of each sample, 375 μL of metabolomics buffer (4:1 K_2_HPO_4_ to KH_2_PO_4_ in dH_2_O at a 0.625 M concentration with a final pH of 7.41 contained an antimicrobial agent of 3.75 mM NaN₃), and 150 mg of zirconium oxide beads were added to a centrifuge tube. Then, samples were homogenized in a Bullet Blender (Next Advance, NY, USA) for 1 min at intensity setting eight, and homogenate was then centrifuged for 5 min at 14,000 g. Next, 365 μL of homogenate and 135 μL of metabolomic buffer were placed in Amicon Ultra 0.5 mL 3 K centrifuge filters and centrifuged for 30 min at 14,000 g. After filtration, 360 μL of the filtrate, 120 μL of metabolomics buffer, and 120 μL of D_2_O with *w*/*v* 0.03% trimethylsilyl propanoic acid (TSP) were moved to a new centrifuge tube. The final dH₂O:D_2_O ratio was 4:1, and the final concentration was 0.5 M for the buffer salts. Finally, samples were centrifuged for 5 min at 12,000 rpm, and 550 μL of supernatant was pipetted to 5 mm NMR tubes for NMR processing.

### 4.4. ^1^H NMR Data Acquisition and Processing

NMR data acquisition used a 700 MHz Bruker Avance III HD spectrometer (Bruker, Milton, ON, Canada) equipped with a TBO-Z probe [66,67,68]. In brief, data were collected using the noesygpprld pulse sequence with these acquisition parameters: number of scans (NS) = 1024, mixing time = 10 ms, spectra window = 20.5 ppm, total number of points (td) = 128 k, total acquisition time = 4.56 s, transmitted offset (olp) = 4.7 ppm, and recycle delay (dl) = 1 s. NMR spectra was automatically phase- and baseline-corrected via TopSpin (v 4.0.6), with the TSP peak used as a 0 ppm reference for chemical shift. The spectra were exported to MATLAB (The MathWorks, Natick, MA, USA) for further processing and statistical analysis. Spectra binning was performed using the dynamic adaptive binning algorithm [69] with manual inspection for any errors. Finally, the binned spectra were Pareto-scaled, normalized to the total unit area of all bins (except for the water peak), and log-transformed.

### 4.5. Statistical Analysis

Univariate and multivariate testing was performed using MATLAB (The MathWorks, Natick, MA, USA) and MetaboanalystR (v 2.0) [70]. A Shapiro–Wilk test was used to determine the normality of the data and the appropriate univariate test. Consistently, a Mann–Whitney U Test (MW) was used, and bins with a *p*-value < 0.05 were considered significant. Bonferroni–Holm correction was applied to correct for multiple comparisons. Supervised multivariant data analysis consisted of variable importance analysis based on random variable combination (VIAVC) [71,72], orthogonal projections to latent structures discriminant analysis (OPLS-DA), receiver operator characteristic (ROC) curves, and predictive accuracy. Multivariate modelling was initially performed on all bins, and then it was only performed for the bins determined to be significant via MW or VIAVC statistical testing. The ROC curve and predictive accuracy were determined only using the bins determined to be significant via VIAVC.

### 4.6. Metabolite Identification and Pathway Analysis

Metabolite identification was performed for the bins that were determined to be significant via either the MW or VIAVC test using Chenomx 8.2 NMR Suite (Chenomx Inc., Edmonton, AB, Canada). Then, identified metabolites were confirmed using the Human Metabolite Database (HMDB) [73]^,^ and only metabolites formerly observed in brain tissue, cerebrospinal fluid, and blood were used. Finally, pathway topology analysis [74] was performed in Metaboanalyst via the *Homo sapiens* KEGG pathway library (v Oct.2019). Relative-betweenness centrality was selected for pathway topology analysis, and the hypergeometric test was used for over-representation analysis.

## 5. Conclusions

The exact biological mechanisms and early predictive markers of AD are still incompletely understood. New biomarker discovery tools such as metabolomics enables the identification of biochemical processes and biomarkers associated with AD for better risk prediction and diagnosis. For the first time, this study uncovered the biochemical pathways associated with AD across eight brain regions. Common metabolites involved in AD pathology across brain regions include amino acids phenylalanine, phosphorylcholine, GABA, NAA, and citric acid, while unique metabolites included pyruvate, lactic acid, succinic acid, myoinositol, and arginine (Table 4). The affected metabolites are part of the biological pathways important for metabolic energy regulation, neurotransmission, and mitochondrial function.

It remains to be confirmed if these changes in cell metabolism reflect pathological or compensatory responses. The extent of metabolic changes is consistent with the neuropathological disease progression of AD and clinical observations of cognitive impairment [4,29]. Moreover, the present study demonstrates that AD leads to metabolic changes in brain regions with less well-documented pathological alterations (PB, DN, and BA24). Therefore, these data provide new directions to better understand the biochemical underpinnings involved in AD. It remains to be determined if these metabolic signatures of AD pathology can be identified in peripheral tissues, such as blood, cerebral spinal fluid, or urine. Nevertheless, the identified metabolic signatures linked to AD may provide potential new avenues for personalized risk prediction or diagnosis.

## Figures and Tables

**Figure 1 ijms-24-14769-f001:**
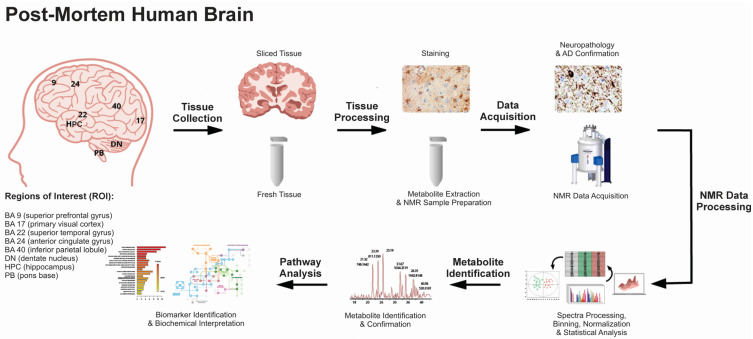
Post-mortem brain tissues (Brodmann areas BA9, BA17, BA22, BA24, BA40, DN, HPC, and PB) were retrieved from the Calgary Brain Bank from categorized control (non-AD) and AD patients. Tissues were processed (metabolic extraction and NMR sample preparation) and analyzed using ^1^H NMR spectroscopy. Both univariate analysis and a multivariate machine learning approach involving permutation testing were used to determine metabolite signatures across the eight brain regions. The results focus on the differences observed in AD tissues compared to non-AD controls. The complete list of metabolites altered in AD tissues compared to controls was used for the pathway topology analysis.

**Figure 2 ijms-24-14769-f002:**
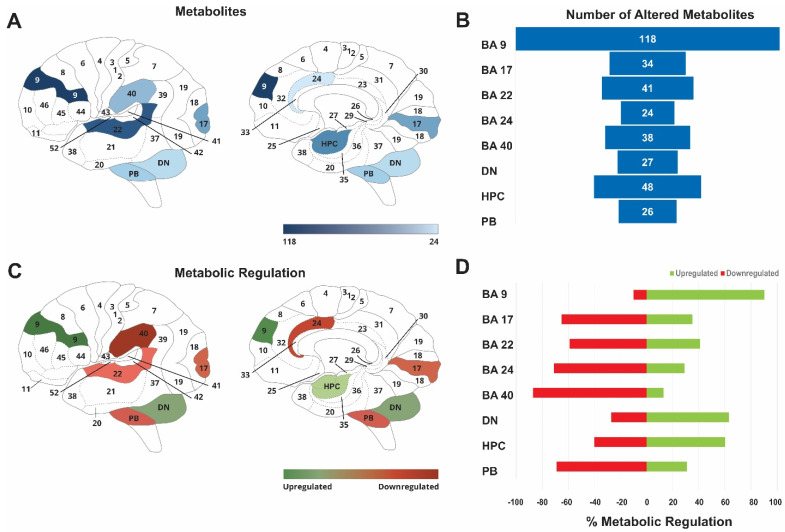
Brain heat maps illustrating the total number of metabolites and regulation changes across brain regions in AD tissues compared to non-AD controls. (**A**) The darkest blue indicates the brain region (BA9) with the most metabolic changes in AD (BA9), while the lighter blue indicates the least number of changes (PB). (**B**) Bar graph showing the actual number of metabolites altered in each region. (**C**) The heat map shows the predominant metabolic regulation changes for each brain region, with red indicating downregulation and green showing upregulation. (**D**) Bar graph indicating % of metabolites upregulated and downregulated in each brain region.

**Figure 3 ijms-24-14769-f003:**
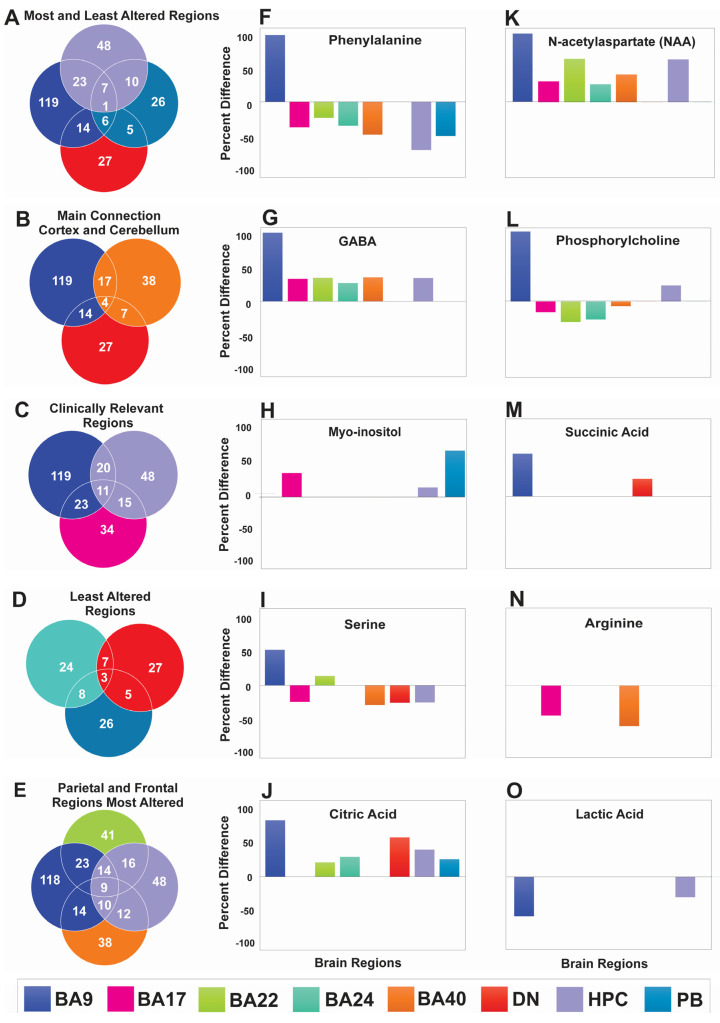
Venn diagrams showing common and unique metabolites across brain regions and specific metabolites identified via VIAVC analysis as biomarkers of AD. (**A**–**E**) Illustrate common metabolites found across various regions that may serve as potential biomarkers of AD. For example, four common metabolites were identified in BA9, BA40, and DN that play roles in signaling between cortex and cerebellum. (**F**–**O**) Illustrate potential biomarkers of AD, as determined via VIAVC best subset analysis. NAA (**K**) and GABA (**H**) were predominately upregulated in all brain regions of AD tissues, while phenylalanine and phosphorylcholine were downregulated.

**Figure 4 ijms-24-14769-f004:**
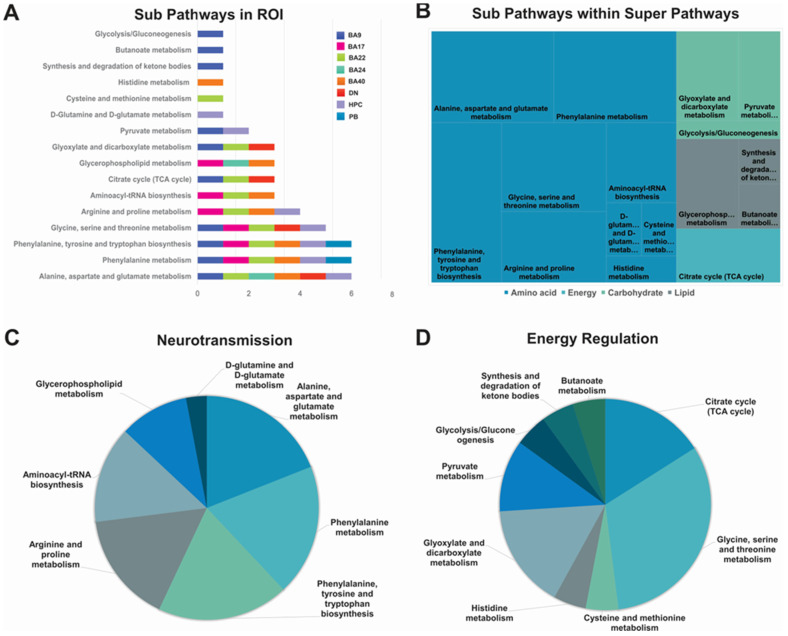
Sub- and super-biochemical pathways involved in neurotransmission and energy regulation throughout brain regions in AD compared to non-AD controls. (**A**) Specific biochemical pathways affected multiple brain regions, with alanine, aspirate, and glutamate metabolism being altered in 6/8 regions. (**B**) Illustration of sub-pathways within super-pathways, showing that most altered metabolites belonged to the amino acid category. (**C**,**D**) Pie diagrams indicating AD-altered pathways involved in neurotransmission and energy regulation.

**Table 1 ijms-24-14769-t001:** Demographics table with experimental groups and brief descriptions. This study involved brain tissues from 11 AD and 11 healthy subject (HS) participants. There was a nearly even split between males and females. The average age of AD participants was 80 years old compared to 65 years for HS. The average neuropathology ABC score was 7.63 out of possible 9 in AD and 1.63/9 in HS subjects.

Experimental Group	Sex	Average Age (yrs.)	PMD (h)	Average Total ABC Score
AD (n = 11)	5 Female; 6 Male	80 ± 9	30 ± 12	7.63/9
HS (n = 11)	5 Female; 6 Male	65 ± 12	41 ± 15	1.63/9

Key: AD, Alzheimer’s disease; HS, healthy subject; PMD, post-mortem delay; ABC Total Score, Amyloid, Braak tau distribution, and CEDAR score, ± upper/lower ranges.

**Table 2 ijms-24-14769-t002:** Summary of statistically significant bins for BA9, BA17, BA22, BA24, BA40, DN, HPC, and PD determined using MW + VIAVC, with corresponding number of metabolites. Q2 and R2Y (and *p* values) from the OPLSDA built from the significant MW + VIAVC bins are also reported.

Brain Region	Number of Significant Bins Out of Total Bins	Corresponding Number of Identified Metabolites	Q^2^ Value	R^2^Y Value
BA9	353/382	118	0.737 (*p* < 5 × 10^4^)	0.835 (*p* < 5 × 10^4^)
BA17	110/364	34	0.736 (*p* < 5 × 10^4^)	0.967 (*p* = 0.003)
BA22	157/380	41	0.738 (*p* < 5 × 10^4^)	0.867 (*p* < 5 × 10^4^)
BA24	61/365	24	0.00601 (*p* = 0.238)	0.727 (*p* = 0.0375)
BA40	84/378	38	0.173 (*p* = 0.1065)	0.798 (*p* = 0.0015)
DN	63/355	27	0.512 (*p* < 5 × 10^4^)	0.749 (*p* = 0.004)
HPC	100/316	48	0.586 (*p* < 5 × 10^4^)	0.77 (*p* = 0.001)
PB	53/299	26	0.624 (*p* < 5 × 10^4^)	0.789 (*p* < 5 × 10^4^)

Note: Q2 and R2Y (and respective *p*-values) from the OPLS-DA built from the significant bins are also reported. For PB, these values are derived from the OPLS-DA constructed from the MW and VIAVC F-ranked bins.

**Table 3 ijms-24-14769-t003:** Summary of number of VIAVC Best Subset bins use to build the ROC for BA9, BA17, BA22, BA24, BA40, DN, HPC, and PD and the predicative accuracy, 95% confidence interval, and AUC for each ROC. Also, the specific metabolites that the VIAVC Best subset bins correspond to is shown.

Brain Region	Number of Bins	Predictive Accuracy	95% Confidence Interval	AUC	Corresponding Metabolites
BA9	24	85.5%	0.827–1	0.976	Niacinamide, Formic acid, FAPy-adenine, Adenosine monophosphate, Oxypurinol Imidazole, Thiamine pyrophosphate, Benzaldehyde, Terephthalic acid, 4-pyrudoxate, Deoxyuridine, L-tryptophan, Nicotinurate, L-tryptophan, Phthalic acid, and L-phenylalanine
BA17	3	98.9%	1–1	1	Glycerol and L-serine
BA22	4	95%	1–1	1	L-phenylalanine and N-acetyl-L-aspartate
BA24	3	94.9%	0.957–1	0.997	2-hydroxy-3-methylvalerate and gamma-aminobutyric acid (GABA)
BA40	6	98%	0.903–1	0.995	Dimethylsulfone, GABA, and Glycerophosphocholine/Sarcosine, Phosphorylcholine, 2-amino-3-phosphonopropionic acid/Arginine/2-Hydroxyvalerate/Leucine
DN	7	81%	0.75–1	0.915	Serine/Glycyl-glycine, Trimethylamine N-oxide, Methylguanidine, Succinic acid, and (R)-3-hydroxybutyric acid
HPC	14	93.8%	0.903–1	0.995	Myo-inositol, Dimethylsulfide, Taurine, L-cystathionine, Selenomethionine, D-lactic acid, and L-lactic acid
PB	2	92.7%	0.901–1	0.993	Citrate and L-isoleucine

**Table 4 ijms-24-14769-t004:** List of possible metabolomic biomarkers in AD identified in post-mortem brain tissues.

Metabolomics Biomarkers of AD
Arginine
Citric Acid
Gamma-Aminobutyric Acid (GABA)
Lactic Acid
Myo-Inositol
N-Acetylaspartate (NAA)
Phenylalanine
Phosphorylcholine
Serine
Succinic Acid

## Data Availability

All data, documentation, and codes used in analyses are available and will be shared upon request.

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
