# Peer review of "Metabolomic Signatures of Alzheimer’s Disease Indicate Brain Region-Specific Neurodegenerative Progression"

_ijms, 2023, doi:10.3390/ijms241914769_

Round 1

Reviewer 1 Report

The work of Ambeskovic et al. concerns a study with a metabolomic approach to define the pathological aspects correlated with Alzheimer. The study highlights the identification of some substrates involved in different biochemical pathways that are increased or reduced. In particular N-acetylaspartate and GABA were upregulated in most regions, suggesting higher inhibitory activity in neural circuits, while other results highlight mitochondrial involvement with energy metabolism. Overall the work is clearly structured although it does not exhaustively support the possible mechanisms underlying the results obtained. Due to the involvement of mitochondria in Alzheimer's pathology, a curiosity that I want to ask concerns whether the authors have carried out analyzes on mitochondrial extracts. Overall, the manuscript can be considered acceptable for publication

I m sorry for the delay into the submission. best regards

Author Response

We welcome and thank the reviewer for the feedback. 

REVIEWER 1:

1. Due to the involvement of mitochondria in Alzheimer's pathology, a curiosity that I want to ask concerns whether the authors have carried out analyzes on mitochondrial extracts.

Thank you for this question. The analysis was not carried out on the mitochondrial extracts. 

Reviewer 2 Report

My suggestions: 

1.  Adding a table for possible biomarkers of metabolomics in AD would be useful.

2, Is it possible that the metabolomic changes could be measured in CSF? Were there any markers in CSF described, which may reflect the brain metabolomic changes? Authors may mention it in the discussion.  

3. Were there any genetic risk factors described for AD, which could be associated with metabolic changes in the brain? The authors may mention it either in the discussion or in the introduction.

4, Is there any link between brain metabolomics and inflammation/oxidative stress? Authors may mention it in the discussion. 

5. The patients, included in the study were early or late onset AD patients? Authors may mention the age of onset of the included patients. 

Reviewer 3 Report

The paper by Metz and colleagues is focused on a study on AD through a metabolomic approach on post-mortem brain specimens. The approach, based on 1H-NMR untargeted metabolomics, is interesting and the paper is well prepared. The paper should be processed further for publications but some improvements are required. I report some comments below.

·         Introduction: the authors are right in observing that many studies are based on targeted metabolomics approach (line 64). They should anyway provide updated references for their statement.

·         Figure 2: panels and fonts are too small and difficult to be read. The artwork should be reorganized (maybe in portrait format?).

·         In the Results section, paragraph 2.2 should not be just a summary of what can be observed in the figures, but rather an explanation of the results with a comment (lines 137-148).

·         The presence of tables and figures in the Materials and Methods section is rather unconventional. I suggest to move them to another part of the manuscript.

·         Figure 4: please check the upper part. I think that the double arrows (pointing up and down at every step) are misleading.

·         Check the title of section 4.4. as there is a typo in 1H-NMR

·         NMR instrument is Bruker Avance and not Advance

Round 2

Reviewer 2 Report

The manuscript is acceptable now.